# Stroke-Related Sarcopenia among Two Different Developing Countries with Diverse Ethnic Backgrounds (Cross-National Study in Egypt and China)

**DOI:** 10.3390/healthcare10112336

**Published:** 2022-11-21

**Authors:** Marwa Mohammed, Jianan Li

**Affiliations:** 1College of Rehabilitation Medicine, Nanjing Medical University, Nanjing 210029, China; 2Faculty of Physical Therapy, Beni-Suef University, Beni-Suef 62521, Egypt; 3First Affiliated Hospital, Nanjing Medical University, Nanjing 210029, China

**Keywords:** age, ethnicity, dyslipidemia, smoking, stroke severity

## Abstract

The prevalence of stroke-related sarcopenia differs according to the diagnostic criteria, the cut-off point for assessment, and ethnicity. Egypt and China are developing countries with different races where no research concerning stroke-related sarcopenia has been performed yet. We aimed to evaluate the prevalence of possible sarcopenia and confirmed sarcopenia among Egyptian and Chinese stroke survivors using the Asian Working Group of Sarcopenia (AWGS-2019) criteria and to assess the variables associated with the prevalence rate. A prospective cross-sectional study was carried out among 200 Egyptian and 195 Chinese stroke survivors from 2019 to 2021 using a structural health questionnaire. A hand-held dynamometer was used to measure grip strength. Anthropometric measures were used to estimate muscle mass. Data were analyzed using SPSS statistics version 20. *p*-values < 0.05 were considered statistically significant. The prevalence of possible sarcopenia ranged from 20.0% to 34.4% among Egyptian and Chinese groups, except for the Egyptian females where it was 52.0%. The prevalence of sarcopenia in both populations ranged from 13.6% to 18.6%. Pre-stroke independent variables that accelerated possible sarcopenia were age, history of dyslipidemia, diabetes mellitus, and ischemic heart disease, but stroke severity was a post-stroke risk factor. Age was the only pre-stroke variable for sarcopenia, while quitting smoking and having good nutritional status were variables for the reduction of possible sarcopenia. Quitting smoking, having a good nutritional status, and early rehabilitation reduced sarcopenia development. Controlling vascular risk factors, enhancing rehabilitation, and nutritional therapy are protective measures against sarcopenia. Longitudinal studies are required to identify further risk factors.

## 1. Introduction

Stroke is the leading cause of disability globally. Post-stroke, skeletal muscles are highly affected in the patients and further contribute to disability and adverse outcomes. Muscle strength, mass loss, and increase in fat mass are observed in the paretic and non-paretic limbs within a similar time course [1,2]. This systemic muscle adaptation is known as stroke-related sarcopenia. Sarcopenia is still a challenging area for stroke patients [3]. It is a predictor of worse functional recovery [4] and it is associated with a fivefold increase in hospital costs compared to patients without sarcopenia [5].

The prevalence of sarcopenia was reported to differ among many studies according to diagnostic methods, characteristics of study populations, and ethnicity. International societies urged for the screening and assessment of sarcopenia in stroke patients with participants from different countries [6]. Unmanaged sarcopenia can rapidly increase the risk of mortality and functional decline. The prevalence of scarce studies identified the prevalence of stroke-related sarcopenia. They were mainly performed in developed countries (Taiwan, South Korea, Japan, and the US) with a wide range from 16.8% to 60.3%, according to confirmed definitions [7,8,9,10,11,12]. These countries had better access to healthcare services. There is no previous research concerning stroke-related sarcopenia in developing countries, especially in populations from different ethnicities with poorly organized healthcare services. It is well known that the global burdens of stroke continuously increase, especially in developing countries [13], leading to an alarming disaster that requires quick intervention.

Egypt and China are two developing countries with diverse ethnical backgrounds; they share stroke characteristics extensively. Stroke is still a vital health concern in Egypt; its prevalence is higher than in other Arabic countries [14,15]. In China, the stroke burden has been exaggerated in the last three decades [16], and is still a challenging public health issue as there is a continuous increase in stroke survivors. Both nations shared a significant rise in stroke risk factors, with a higher prevalence in men than women and a higher prevalence of ischemic stroke rather than hemorrhagic stroke [17,18].

Previous studies showed that stroke-related sarcopenia is very common in stroke patients [8,9] and is not age-dependent [7]. The etiology underlying this high prevalence is still less clear [11]. It reveals features that distinguish it from sarcopenia observed in ageing, which is the rapid decline of muscle mass [19]. Despite multiple studies concerning stroke-related sarcopenia [7,8,9,11], none identified these factors. Sarcopenia in stroke might be induced due to pre-stroke risk factors or caused by stroke comorbidities. There is still overlap, and it can be difficult to distinguish these factors, especially in patients in the acute phase. The acute phase often leads to sudden physical disability and loss of consciousness. As a result, identifying these variables is of interest because they may be modifiable and could contribute to better management.

There are several diagnostic criteria for sarcopenia with different cut-off definitions [20,21,22,23]. Still, these criteria are based mainly on muscle mass as a core for sarcopenia assessment, in addition to handgrip strength (HGS), and/or gait speed. The new updates of the AWGS-2019 focus on muscle strength as a critical characteristic of sarcopenia [11]. Muscle strength is better than mass in predicting adverse outcomes [24,25,26,27]. The AWGS-2019 diagnostic criteria propose the concept of possible sarcopenia for early detection and treatment of stroke survivors with possible sarcopenia to prevent further muscle loss. Therefore, detecting possible sarcopenia and sarcopenia in the early stages is crucial. It may significantly contribute to less morbidity and mortality related to the condition. We hypothesize that the prevalence of possible sarcopenia and sarcopenia will be higher than presented in developed countries. Therefore, in this study, we aimed to:(1) evaluate the prevalence of possible sarcopenia and sarcopenia using the AWGS-2019 criteria among Egyptian and Chinese stroke survivors in outpatient department services; (2) identify pre- and post-stroke contributing variables that accelerate or decline the prevalence of possible sarcopenia and sarcopenia among both populations.

## 2. Materials and Methods

### 2.1. Ethics Approval

This study had been reviewed and approved by the Human Research Ethics Committee of the First Affiliated Hospital of Nanjing Medical University, Jiangsu province, China (approval ID: 2018. SR.0024), and an ethical committee from the Faculty of Physical Therapy, Beni-Suef University, Egypt(approval ID: BSUPT/10/11/2020). All participating subjects (Egyptian and Chinese stroke survivors) had received a verbal explanation and written detailed information on the study and signed consent forms for the interview. The processing of sensitive personal data was based on the protocol of following the ethical principles of the Helsinki declaration.

### 2.2. Respondent

The out-patient cross-sectional study held between October 2018 and December 2021 considered study populations within the age 30–80 years. They had a stroke less than two months ago, either ischemic or hemorrhagic stroke, confirmed by CT or MRI scans.

The basic characteristics of the population, besides the inclusion and exclusion criteria of the study populations, are presented in Figure 1.

### 2.3. Method

The structural questionnaires were developed which included: (1) basicdemographical information of the two study populations; (2) clinical data included: stroke type, age, sex, time since stroke onset, comorbidities using the guidelines of Charlsoncomorbidity index (CCI), and stroke severity was assessed by using National Institutes of Health Stroke Scale (NIHSS) on the first day the patient came to the out-patient department of rehabilitation and physical therapy by a trained physician.

Assessment if the patient was illegible for inclusion criteria, the following evaluation was done within 48h. Dependence activities of daily living (the modified Rankin scale [mRS]), nutritional status (the Mini Nutritional Assessment-Short Form [MNA-SF]), physical and cognitive function (functional independent measure [FIM]), weight, height, handgrip strength [HG], and calf circumferences [CC] were collected by the same physiotherapist in Egypt and China.

The structural questionnaires were written in English and then translated into Chinese for Chinese populations and translated to Arabic for use among Egyptian populations.

### 2.4. Inclusion and Exclusion Criteria of the Study Populations

#### 2.4.1. Inclusion Criteria

The inclusion criteria of the patients included: (1) age range from 30 to 80years who had a stroke less than two months ago with evidence of ischemic or hemorrhagic stroke confirmed by CT or MRI scans; (2) following patients entered with a stable medical condition, consciousness, and were cooperative with the first-ever onset of stroke; and (3) cases were ambulant before the stroke incident.

#### 2.4.2. Exclusion Criteria

The main exclusion criteria among the study population included: (1) patients with major concurrent illness (renal failure, severe heart failure, or end-stage organ disease), followed by; (2) patients with disturbed consciousness, lower limb amputation, an implanted pacemaker, or altered hydration conditions such as oedema.

#### 2.4.3. Sample Size Calculation

The sample size was calculated using Epi-info7 based on results in the previous study [12].With an expected frequency of 53%. A sample size of at least 181 contributors would be required separately in each ethnic group with a power of 95% and an alpha error of 0.05.

### 2.5. The Recruiting Criteria

#### 2.5.1. Measurements of the Study Populations

##### Definition

We used the Asian Working Group for Sarcopenia2019 (AWGS-2019) algorithm for the sarcopenia definition in the Egyptian and Chinese groups [11].AWGS-2019 used the sequence based on case-finding, the presence of possible sarcopenia, and sarcopenia [11]. Despite the gait assessment used to diagnose sarcopenia severity, we did not include it as the applicability of gait speed assessment may be limited in stroke patients.

#### 2.5.2. Case Finding (Screening for Sarcopenia)

AWGS-2019 recommends the SARC-F questionnaire to produce a self-report assessment of sarcopenia signs. Its scores range from 0 to 10, where a score≥4 was defined as having a risk of sarcopenia [28]. The accuracy of SARC-F for sarcopenia was 65.8%, with a sensitivity and specificity of 62.3% and 71.4%, respectively, using AWGS-2019 [29]. It has a low-to-moderate sensitivity (47.4%) and a very high specificity (87.3%) to predict low muscle strength [30].It has excellent inter-rater and test-retest reliability with an intra-class correlation coefficient (ICC) of 0.90 (95% CI 0.76–0.96) and 0.86 (95% CI 0.66–0.94), respectively [31].

#### 2.5.3. Muscle Strength Assessment (Possible Sarcopenia)

Measuring HGS is simple, reliable, and correlates moderately with strength in other body compartments. For both groups, it was measured by (SPGHOME strength measurement meter, digital hand dynamometer) with elbows flexed to 90 and forearms in a neutral position. The applicants were educated to grip the device as much as possible in the non-affected hand, it was tested three times, and the highest kilograms were recorded. It has excellent test-retest reliability ICC of (0.85) [32]. HGS was <28.0 kg for men and <18.0 kg for women [11].

#### 2.5.4. Muscle Mass Assessment for Confirmed Sarcopenia

Although computed tomography (CT), magnetic resonance imaging (MRI), dual-energy X-ray absorptiometry (DXA), and bioelectrical impedance analysis (BIA) have all been used to estimate skeletal muscle index (SMI) for the identification of sarcopenia, these tools are not yet widely implemented in everyday clinical practice because of a lack of portability. It is too expensive to screen for sarcopenia, especially in developing countries. No definitive tool has been used to determine cut-off values for muscle mass and sarcopenia in stroke patients [19]. Additionally, there is still a lack of cost-effectiveness studies on sarcopenia screening in clinical studies [6].Therefore, in our research, anthropometry is used as an alternative to estimate appendicular skeletal muscle (ASM). Anthropometry is non-invasive, inexpensive, and can be applied in clinical and research settings [33].

In both groups, we used the Santos equation $ASM=-0.029\times age\left(years\right)+7.523\times sex+ethinicity+0.768\times calf\;circumferences\left(cm\right)-10.427\;$. For sex, 0 is used for women and 1 for men, ethnicity = (−0.402) [34]. This formula was made using data from the NHANES study between 1999 and 2006. The calf circumference was calculated from a non-affected limb to eliminate the effect of stroke. This equation estimated ASM in all age groups and explained almost 90% of the DEXA-measured ASM variability (adjusted R^2^ = 0.88; root mean square error = 1.95 kg), with Lin’s concordance correlation coefficient and Bland–Altman’s >90% [34]. Its validity using the bootstrap-corrected root mean square error was 1.91 [35].The cut-off of the SMI was <7 kg in males and <5.4 kg in females used in this study, according to AWGS-2019.

### 2.6. Statistical Methods

Data processing and analysis used IBM SPSS Statistics 20 (SPSS Inc., Chicago, IL, USA). Descriptive statistics were analyzed, and results were reported as mean ± standard deviation (SD) for parametric data and Median ±interquartile range (IQR) for non-parametric data. Statistical assumptions were verified through tests of normality by the Kolmogorov–Smirnov test. Between-group comparisons for those with and without sarcopenia were made using the t-test for parametric standard distribution variables, the Mann–Whitney U test for non-parametric continuous variables, and the Chi-square test which had been applied for comparing different groups when variables were categorical. Spearman correlation tests were performed to detect correlation coefficients between pre-stroke and post-stroke factors contributing to possible sarcopenia (HGS) and sarcopenia (SMI) among Egyptian and Chinese groups. Multivariate regression stepwise analysis was used to identify pre-stroke and post-stroke risk variables for possible sarcopenia and sarcopenia. All statistical tests at the level of statistical significance were set at *p*-values < 0.05 and were considered statistically significant.

## 3. Results

A total of 395 participants, including patients (200 Egyptian and 195 Chinese), were investigated, and (125 Egyptian and 109 Chinese) invalid data were eliminated. The response rate of the study was 61.5% Egyptian and 64.14% Chinese.

The basic characteristics of the study population are listed in Table 1—characteristics of stroke survivors between the male Egyptian versus Chinese patients. The study included 395 patients (200 Egyptian and 195 Chinese). Among the Egyptians, there were a total of 125(62.5%) male and 75(37.5%) female patients, while among the Chinese, there were a total of 125(62.5%) male and 70(35.9%) female patients. There was a significant difference in dyslipidemia, SMI, size of affected and non-affected calf-FIM motor, and FIM total reported among Chinese and Egyptian (*p* < 0.05).

Characteristics of stroke survivors between those with and those without sarcopenia among Egyptian and Chinese females are illustrated in Table 2. A total of 75 Egyptian and 70 Chinese females were included. About 26(34.6%) Egyptian and 10(14.2%) Chinese females had low SMI. There were significant differences in height, SMI, and size of affected and non-affected calf reported among Egyptian and Chinese females (*p* < 0.05).

The prevalence of possible sarcopenia and sarcopenia among Egyptian and Chinese groups according to AWGS-2019 is reported in Figure 2 and Figure 3. The prevalence of possible sarcopenia among Egyptian males and females was 34.4% and 52.0%, respectively, while it was 20.0% and 24.2%among Chinese males and females, respectively. The prevalence of sarcopenia was 16.0% and 18.6 % among Egyptian males and females, respectively, while 13.6% and 14.2% among Chinese males and females, respectively.

The Spearman correlation between the pre- and post-stroke contributing variables that increase or decrease HGS and SMI among Egyptian and Chinese populations is reported in Table 3.

Results demonstrated that age and dyslipidemia were common pre-stroke variables that negatively correlate with HGS in the Egyptian and Chinese groups. In contrast, the history of smoking positively correlates with HGS. The common post-stroke variables that positively correlate with HGS were FIM motor and FIM cognitive in addition to MNA-SF, while stroke severity negatively correlates with HGS. The common pre-stroke variable that positively correlated with SMI was a smoking history with no common pre-stroke contributing variables accelerating SMI loss. While common post-stroke contributing variables positively associated with SMI was a good nutritional status (MNA-SF) with no common post-stroke contributing variables accelerating SMI loss.

The multivariate regression stepwise analysis identifies pre- and post-stroke contributing variables of possible sarcopenia and sarcopenia in both populations explained in Table 4. The multivariate analysis showed that pre-stroke comorbidities contributing to the increased prevalence of possible sarcopenia among the Egyptian group were age, history of DM, IHD, and dyslipidemia. Any increase in age or having a history of DM, IHD, or dyslipidemia cause a decline in muscle strength by 25%, 19.8%, 19.2%, or 21.3%, respectively. While in the Chinese group, the pre-stroke comorbidities contributing to the increased prevalence of possible sarcopenia were age and dyslipidemia. Any increase in age by one year or having a history of dyslipidemia caused a decline in muscle strength by 29% and 22.5%, respectively.

On the contrary, smoking history was an independent pre-stroke variable that declined the prevalence of possible sarcopenia among both Egyptian and Chinese groups. Quitting smoking enhanced muscle strength by 30.3% and 29.3% among Egyptian and Chinese groups, respectively. The common post-stroke variable that declined possible sarcopenia among Egyptian and Chinese patients was good nutritional status, as good nutritional status enhanced muscle strength by 31.9% and 58.6% in both groups, respectively. While stroke severity using the NIHSS scale was an independent variable for possible sarcopenia in the Egyptian group, the stroke severity decreased muscle strength by 40.8%.

The pre-stroke variable that increased the prevalence of sarcopenia among the Egyptian group was age, as any increase in age by one year caused muscle mass decline by 27.1%. In comparison, the history of smoking was an independent variable for a decrease in the prevalence of sarcopenia by 46.2%. The post-stroke factor that declined the prevalence of sarcopenia among the Egyptian group was rehabilitation history in the acute phase, which increased muscle mass by 14.7%, and good nutritional status (MNA-SF) enhanced increased muscle mass by 30.3%. At the same time, only MNA-SF in the Chinese group improved muscle mass by 24.4%.

## 4. Discussion

This cross-sectional study investigated the prevalence of possible sarcopenia and sarcopenia among two diverse ethnic groups (Egyptian and Chinese stroke survivors) using AWGS-2019 and pre- and post-stroke variables for probable and confirmed sarcopenia.

Our results reveal that the prevalence of possible sarcopenia using AWGS-2019 ranges from 20.0% to 34.4% among Egyptian and Chinese groups, except for the Egyptian females (52.0%). Our results matched the only study that identified the prevalence of possible sarcopenia in stroke survivors as 32.1% using EWGSOP2 in chronic stroke patients admitted to inpatient rehabilitation [36]. In our study, we measured HGS on the non-affected side as most of the daily activities are expected to be carried out by it. Weakness of this side may lead to increased physical disability, hospitalization, falls, and, in turn, affect ADL performance. This side is always neglected in rehabilitation; it is critically important to include the non-affected side in a rehabilitation program. According to previous studies, weakness starts to develop in the non-affected side within two days of stroke onset [37]. Earlier studies showed that strengthening exercises of both the paretic and non-paretic sides is principal in the functional treatment of stroke [38]. There should be a greater focus on the early assessment and treatment of stroke patients with possible sarcopenia, especially in young survivors, in order to prevent the development of sarcopenia.

The high prevalence of possible sarcopenia in Egyptian females was 52.0%, which may be attributed to a sedentary lifestyle and lack of physical activity in Egyptian culture, which makes their muscles weaker than those of Chinese females who walk much more, protecting them against sarcopenia [39]. As stroke survivors are physically inactive, early mobilization is required to prevent worsening the condition.

The prevalence of sarcopenia in both populations ranged from 13.6–18.6%. Our results matched the results of previous studies [7,36]. Additionally, our results were much lower than those mentioned in developed countries where the prevalence was 53.5% [8], 39.7% [9], 48.3% [10], and 53.6% [12].This is unexpected to our hypothesis. This significant difference between our results and previous ones may be due to multiple causes. Our participants were recruited from the outpatient department, where the patients had less severe conditions than patients in convalescent rehabilitation wards [8,10,12]. Furthermore, their mean age was lower by about two decades, and they are in a chronic phase where they may be ambulant or well-nourished. Wherever the chance of neurological recovery occurred participants in the acute stage reported in this study [8,10,12]. Our study used an anthropometric equation to assess muscle mass despite the low accuracy of anthropometric measures to detect muscle mass. The equation used in our research is mainly based on calf circumferences. The calf circumferences used in the equation measured in the non-affected calf of our participants were (31.53+2.1) (31.18+2.98) among males and (31.4+2.1) (29.1+2.22) among females with sarcopenia. It was lower than the suggested cut-off values of calf circumference for predicting low muscle mass which are:<34 cm in men and <33 cm in women [40].Previous studies reported that calf circumference was positively correlated with DXA-measured ASM and SMI [40]; additionally, CC could be used as a surrogate marker of muscle mass for diagnosing sarcopenia [40]. It may be a valid option for assessing sarcopenia in stroke survivors without gold-standard methods; prospective studies are required to illustrate this. In stroke survivors, sarcopenia prevalence may vary according to multiple factors, including stroke stage, severity, age, physical activity level before the stroke, comorbidities, and nutritional and rehabilitation status; in addition to diagnostic criteria, the cut-off point used in assessment, the race that occurs in sarcopenia with ageing. Further studies are required to identify the impact of these variables on sarcopenia prevalence in stroke patients.

Although Egypt and China have diverse racial and genetic backgrounds, they have some similarities and differences.

They shared some pre-stroke variables that induce possible sarcopenia (age and dyslipidemia). Our results revealed that age is the common pre-stroke variable independently associated with increased prevalence of possible sarcopenia in whole study groups. Age was significantly higher in stroke patients with possible sarcopenia (*p* = 0.006) [36]. Our results matched previous studies that showed there was a negative relationship between age and grip strength [41,42]. The combination of age and stroke enhances muscle weakness more rapidly, even at young ages, when compared tothe normal declines that occur with chronological age. A previous study illustrated that possible sarcopenia was common with ageing, where its prevalence among elderly 60–94 y was 38.5% [43]. Despite this quiet matching with our results, our participants had a younger mean age ±SD of 55.54 ± 13.1 and 52.8 ± 15.15 y than reported in this study, thus illustrating that stroke enhances the occurrence of possible sarcopenia three decades earlier than induced by age. Enhancement of a positive and healthy lifestyle contributes to the improved physical performance of older people [44].

Our results illustrated that dyslipidemia was an independent variable for possible sarcopenia. Studies from west to east demonstrated that relative HGS showed a significant inverse association with dyslipidemia [45,46,47]. Matching our results, previous studies illustrated a positive correlation between HGS and healthy lipids, e.g., high-density lipoprotein (HDL) [45,46,47]. Because the risk factors of dyslipidemia continue to increase due to changes in several lifestyle habits, including a sedentary lifestyle, cigarette smoking, diet-foods rich in saturated foods and trans fat, prolonged elevation of insulin levels, excessive alcohol consumption, and type 2 diabetes, it is necessary to enhance the awareness, treatment, and control of dyslipidemia in both nations. Since a large percentage of our survivors had a history of dyslipidemia, monitoring their lipid profiles is vital. The long-term effects of uncontrolled dyslipidemia on muscle mass loss have not yet been identified; further studies are required. Dyslipidemia is a risk factor for cardiovascular disease and stroke. Earlier studies confirmed that transient ischemic attack (TIA) or ischemic stroke survivors are more vulnerable to recurrent atherosclerotic cardiovascular diseases (ASCVD) [48]. Therefore, they should receive pharmacologic treatment to reduce their risk of stroke recurrence, myocardial infarction (MI), and vascular death [48]. The American Heart Association (AHA) urges all adults to follow a healthy diet, engage in regular exercise, and avoid smoking [49]. Previous studies have demonstrated the protective properties of muscle strength in adults with cardiometabolic risk factors [50,51]. In addition, muscle strength improves triglycerides, body mass index (BMI), and high-density lipoprotein (HDL) levels [52]. The combined effect of pharmacological therapy and strengthening exercises on HGS, LDL, and HDL levels in stroke survivors has not yet been identified.

The Egyptian group was superior in the number of variables (age, history of DM, IHD, and dyslipidemia) that induce possible sarcopenia. The accumulation of these multiple variables explains the variance in HGS among the Egyptian and Chinese groups, which were 23.32 ± 17.7 and 29.2 ± 9.34, respectively. Similar to our results, according to a nationwide survey of Korean adults, relative HGS was significantly inversely related to cardiovascular risk factors, particularly low HDL, ageing, DM, and hypertension [46]. Previous studies illustrated that the prevalence rate of sarcopenia is higher in elderly patients with cardiovascular disease (CVD) compared to the general population [53]. Muscle strength was five times weaker among patients with diabetes than normal [51]. The high prevalence of sarcopenia in people with diabetes is two to three times more than in controls [54,55]. Stroke patients are physically inactive, which only accelerates the development of sarcopenia. Rapid management of stroke risk factors is required as sarcopenia is related to increased mortality risk and reduced quality of life [53]. As muscle strength loss occurs before muscle mass loss and the accumulation of multiple risk factors, cohort studies are warranted to identify how quickly sarcopenia will develop among those survivors.

Our results demonstrate that stroke severity measured by the NIHSS is an independent post-stroke variable increasing the prevalence of possible sarcopenia among the Egyptian group. However, the NIHSS assesses stroke severity and records stroke patients’ performance. It does not measure the muscle tone of the affected side nor its relationship with muscle mass, strength, or functional outcomes. About a third of stroke survivors suffer from spasticity [56]. Spastic muscles also suffer from changes in architecture and quality that can contribute to lower muscle strength [57], and hypoactivity caused by the stroke can decrease muscle mass [57]. Previous studies illustrated that muscle mass loss is sustained on the paretic side during the first year in severely impaired patients with a significant increase in fat mass [58]. Although there was no association between stroke severity and muscle mass loss in our study, muscle mass loss is expected to occur with time. The impact of time since stroke could be a factor for muscle mass loss; therefore, longitudinal studies to follow up on body composition are required.

Our study reported that smoking history was a common independent variable for decreasing possible sarcopenia among the Egyptian and Chinese groups. It is also a variable for the decline in the prevalence of sarcopenia in the Egyptian group only. Although most of our survivors had a smoking history, none currently smoke. Matching our results, earlier studies illustrated that smoking decreases muscle strength and is associated with an accelerated decline of HGS [59]. In an animal study following six months of cigarette smoking exposure, there was a significant fiber-type shift from IIa to IIx/b and a reduction in muscle force. However, these studies were not performed on human beings [60]. Regardless, previous studies confirmed that tobacco smoking is a risk factor for sarcopenia [61,62], as cigarette smoking is linked with low physical activity levels and impaired nutrition [63]. Studies confirmed that smoking is a reversible risk factor for sarcopenia [64]. Our results showed that cigarette abstinence, normal nutritional status, and early rehabilitation of stroke patients in the acute phase had an anabolic effect on skeletal muscle in a short period. Our results matched the previous study’s results that cigarette abstinence positively impacts body composition in post-menopausal women [65]. When confounding factors such as caloric intake and physical activity were controlled, there were significant increases in body weight, fat mass, muscle mass, and functional muscle mass [66].

Our results revealed that age was only an independent variable for sarcopenia in the Egyptian group. Our results matched a previous study where pre-stroke sarcopenia was significant in elderly patients with acute stroke [65]. Pre-stroke sarcopenia is associated with poor functional outcomes and with prolonged hospital stays when compared topre-stroke non-sarcopenic patients [65]. Since age is a significant risk factor for sarcopenia [67] and a dominant risk factor for stroke [68], when stroke survivors get older, muscular function in both limbs may decline at greater rates than in individuals without stroke [19]. It is critical to assess sarcopenia periodically in the elderly as stroke and sarcopenia share a common risk factor that may influence each other intensely, especially in societies with elevated lifespans and dense populations. As a result, the motivation of the elderly to increase their physical activity could be a preventive measure against sarcopenia. Physical exercise in progressive resistance exercise is the fundamental therapeutic strategy to prevent and reverse sarcopenia [69]. Exercise prescription for fragile elderly patients with stroke is warranted.

Our results demonstrated that good nutritional status was a similar post-stroke factor that decreased the prevalence of possible sarcopenia among Egyptian and Chinese stroke survivors. The previous study illustrated that poor oral status was associated with sarcopenia, reduced muscle mass, and reduced strength in post-acute stroke patients [8]. In our study, less than half of our participants had normal nutrition, no swallowing difficulties, and normal weight and BMI in both nations. This indicates that normal food with no swallowing difficulties protects against muscle strength loss and muscle mass loss post-stroke. Our results confirmed a growing body of evidence regarding survival benefits for overweight and obesity in stroke patients [70,71]. Results from patients with an acute stroke or transient ischemic attack revealed that the mortality risk was lower in overweight patients [hazard ratio (HR): 0.69, 95%CI: 0.56–0.86], and most lacking in obese and very obese patients [HR: 0.50, 95%CI: 0.35–0.71 and HR: 0.36, 95%CI: 0.20–0.66, respectively] compared with normal body mass indexes [70].In contrast, about a third of our participants were at risk of malnourishment. Caution should be taken with regular assessment to prevent sarcopenia. More focus on the nutritional status of stroke survivors is warranted as rehabilitation outcomes are poor in malnourished stroke patients [72].

We observed that rehabilitation during the hospitalization period in the acute phase in the Egyptian group was only an independent variable that protects against muscle mass loss. This improvement may vary according to hospitalization period, swallowing disorder or not, stroke severity, age, and nutritional status of the patient. Further studies are required to illustrate this variability. Our results confirmed with earlier studies that rehabilitation nutrition improves physical and mental function, ADLs, and quality of life(QOL) of the elderly with disabilities [73]. Minor studies included rehabilitation nutrition in stroke patients. The previous study showed an eight-week program of leucine-enriched amino acid supplementation, besides low-intensity resistance training, increased muscle mass, strength, and physical function in post-stroke patients with sarcopenia [74].

Strengths of our study:(1) This is the first study to compare stroke-related sarcopenia in two different ethnic groups, only Hans and Egyptian populations, with a relatively large sample size. (2) Additionally, it is the first study to identify the prevalence and contributing factors of possible and confirmed sarcopenia among stroke survivors with ischemic and hemorrhagic strokes.

Limitations of our study:(1) A single hospital study in Egypt and China is hard to generalize to all Egyptian or Chinese populations. (2) The apparent percentage of sarcopenia in our study is lower than expected because our Egyptian group was from Cairo, the capital city of Egypt. The Chinese group was from Nanjing, the capital city of Jiangsu. Both groups had a high socioeconomic status. Better access to healthcare systems in Lower Egypt or rural regions in China requires more prospective studies for stroke patients in different areas in Egypt and China. (3) We measured SMI based on predicted equations validated for both populations, which may carry a basis of overestimation or underestimation. Further studies are required to confirm our results using SMI using DEXA or CT scans. (4) All of our participants are limited to patients admitted to outpatient services, which could have less severe disease than those in-home settings or acute phases. (5) It is a cross-sectional study; causality between sarcopenia and its contributing factors could not be speculated and additional longitudinal studies are required.

## 5. Conclusions

Several variables play a significant role in developing possible sarcopenia and sarcopenia, and these variables could be modified. Enhancing awareness, treatment, and control of risk factors in both nations is essential. There must be increased focus on the importance of lifestyle modifications, healthy diet and exercise, stopping smoking, and medical treatment to control stroke risk factors. Stroke survivors face multiple barriers during their post-stroke life and require multidisciplinary rehabilitation and cooperation of all medical teams during their journey post-stroke. Home-based exercises should educate patients’ relatives to maintain flexibility and prevent muscle mass loss. Periodic assessments of sarcopenia in a stroke survivor with a severe case and a risk of malnutrition determines how quickly the condition develops. Besides studying sarcopenia in patients with recurrent stroke, stroke recurrence is more common.

## Figures and Tables

**Figure 1 healthcare-10-02336-f001:**
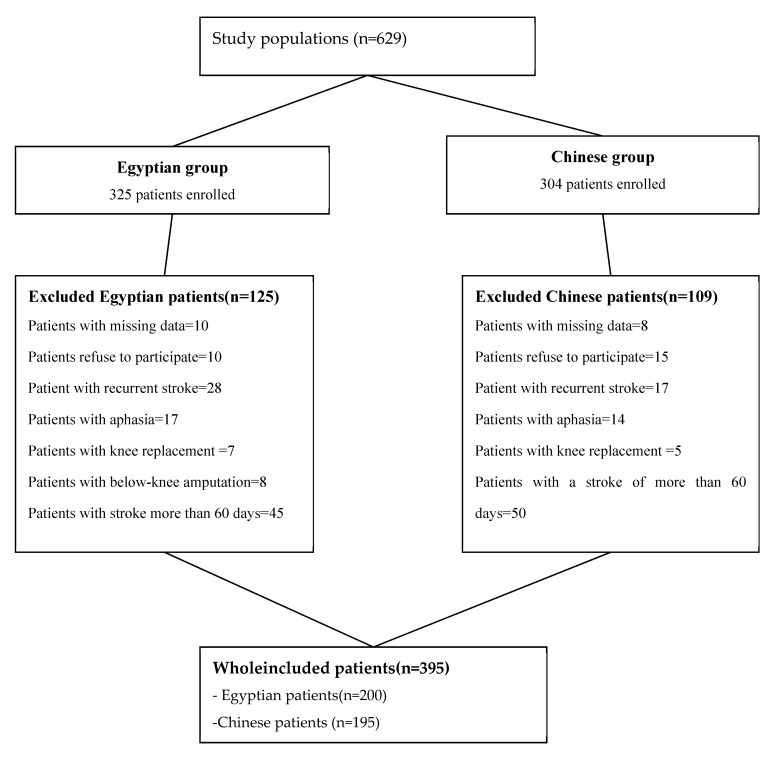
Flowchart of study populations.

**Figure 2 healthcare-10-02336-f002:**
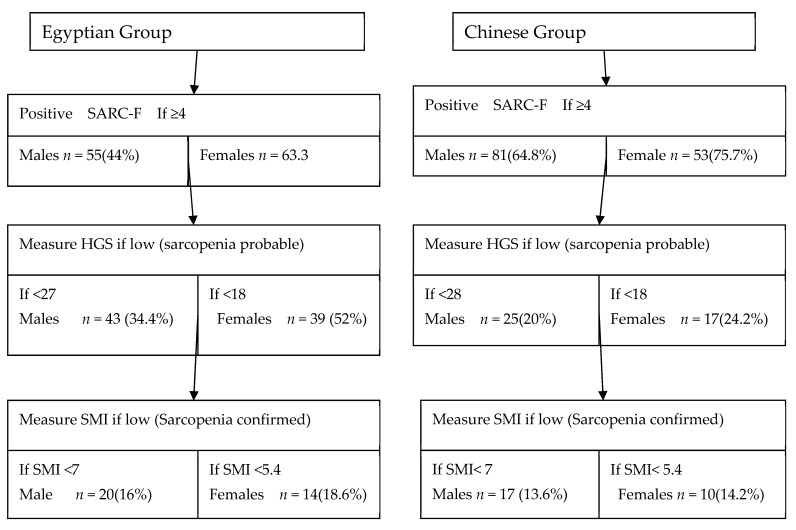
Prevalence of stroke-related sarcopenia in both Egyptian and Chinese populations using AWGS-2019.

**Figure 3 healthcare-10-02336-f003:**
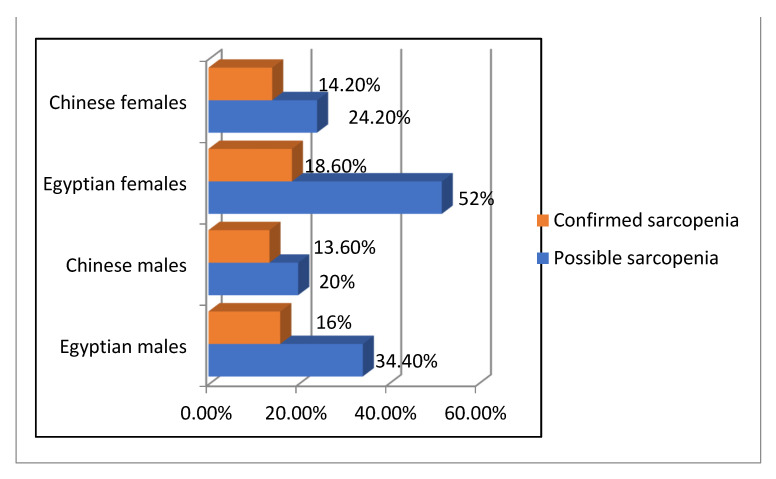
Illustrates the prevalence of possible and confirmed sarcopenia among Egyptian and Chinese patients.

**Table 1 healthcare-10-02336-t001:** The basic characteristic of stroke survivors between the male Egyptian versus Chinese patients with or without sarcopenia.

Variables		Stroke Survivors in EgyptianPatients				Stroke Survivors in Chinese Patients			
		Total, n (%)	Without Sarcopenic	With Sarcopenic	*p*-Value	Total	Without Sarcopenic	With Sarcopenic	*p*-Value
Gender		200				195			
	Males	125(62.50)	99	26		125(64.10)	98	27	
	Females	75(37.50)				70(35.90)			
Age mean ± SD		55.54 ± 13.1	51.6 ± 11.11	67.1 ± 7.30	0.000 ^c^	52.8 ± 15.15	53.09 ± 15.07	53.40 ± 15.25	0.925 ^c^
Stroke type, n (%)									
	Ischemic	121(60.50)	56(56.60)	18(69.20)	0.242 ^a^	84(43.10)	42(42.9)	15(55.6)	0.241 ^a^
	Hemorrhagic	79(39.50)	43(43.40)	8(30.80)		111(56.9)	56(57.1)	12(44.4)	
Affected side, n (%)									
	Right side Left side	102(51) 98(49)	42(42.40) 57(57.60)	18(69.20) 8(30.80)	0.015 ^a^	66(32.80) 129(66.20)	39(39.80) 59(60.20)	9(33.30) 18(66.70)	0.541 ^a^
Smoking n (%)									
	No	110(55)	27(27.30)	9(34.60)	0.462 ^a^	120(61.50)	48(49)	20(70.4)	0.020 ^a^
	Yes	90(45)	72(72.70)	17(65.40)		75(38.50)	50(51)	7(25.9)	
Drinking history, n (%)									
	No	200(100%)	99(100)	26(100)	------	99(50.70)	35(35.70)	13(48.1)	0.240 ^a^
	Yes	0(0.0)	0(0.0)	0(0.0)		96(49.20)	63(64.30)	14(51.90)	
HTN, n (%)									
	No	29(14.50)	17(17.20)	0(0.0)	0.023 ^a^	49(24.70)	20(20.40)	5(18.50)	0845 ^a^
	Yes	171(85.5%)	82(82.8%)	26(100)		146(73.7%)	78(79.20)	22(81.5)	
DM, n (%)									
	No	118(59)	67(67.70)	12(50)	0.095 ^a^	24(12.30)	13(13.30)	3(11.10)	0.767 ^a^
	Yes	82(41)	32(32.30)	12(50)		171(87.70)	85(86.70)	24(88.90)	
IHD, n (%)									
	No	142(71)	81(81.80)	12(46.20)	0.000 ^a^	165(84.60)	79(80.60)	22(81.50)	0.919 ^a^
	Yes	58(29)	18(18.20)	14(53.80)		30(15.40)	19(19.40)	5(18.50)	
Performed conventional rehabilitation in the early phase, n (%)									
	No	116(58)	53(53.50)	15(57.70)	0.705 ^a^	65(33.30)	35(35.70)	14(51.90)	0.128 ^a^
	Yes	84(42)	46(46.50)	11(42.30)		130(66.70)	63(64.30)	13(48.10)	
Dyslipidemia									
	No	140(70)	76(76.80)	12(46.2)	0.002 ^a^	72(36.90)	66(67.0)	11(40.70)	0.012 ^a^
	Yes	60(30)	23(23.20)	14(53.8)		123(63.10)	32(32.70)	16(59.0)	
Mini Nutritional Assessment-Short Form(MNA-SF)									
	Malnutrition	59(29.50)	12(12.10)	17(65.40)	0.000 ^a^	65(33.30)	24(24.50)	10(37)	0.297 ^a^
	Risky	59(29.50)	33(32.30)	4(19.20)		45(23.07)	22(22.40)	7(25.9)	
	Normal	82(41.0)	55(55.60)	4(15.40)		85(43.50)	52(53.10)	10(37)	
FIM motor, median, IQR									
	FIM motor	48(45–72)	46(56–75)	38(41–59)	0.000 ^b^	55(40–66)	52(48–70)	52(40–65)	0.040 ^b^
	FIM cognitive	31(28–33)	32(30–34)	24(23–29)	0.000 ^b^	31(29–33)	31(30–33)	20(28–33)	0.207 ^b^
	FIM total	81(72–105)	98(87–108)	51(65–89)	0.000 ^b^	62(69–100)	61(77–101)	60(67–96)	0.037 ^b^
Modified Rankin scale		2(1–4)	2(1–3)	4(2–5)	0.000 ^b^	4(3–4)	4(3–4)	4(3–4)	0.288 ^b^
Time from stroke onset per month, median, IQR		57(34–43)	47(36–44)	35(35–43)	0.400 ^b^	49(35–44)	36(36–45)	30(38–46)	0.153 ^b^
Weight/kg mean ± SD		71.62 ± 11.0	77.01 ± 8.50	63.9 ± 8.70	0.000 ^c^	69.9 ± 12.12	74.88 ± 10.10	76.92 ± 11.07	0.365 ^c^
Height/m		167.60 ± 8.1	172.03 ± 5.38	171.79 ± 5.60	0.779 ^c^	170.0 ± 7.26	173.76 ± 3.7	176.4 ± 2.6	0.001 ^c^
BMI (kg/m2)		25.3 ± 4.20	26.33 ± 3.30	21.01 ± 2.80	0.000 ^c^	24.0 ± 3.04	24.7 ± 2.9	24.3 ± 3.40	0.902 ^c^
SMI (kg/m2)		7.1 ± 1.50	8.3 ± 0.822	6.2 ± 0.45	0.000 ^c^	7.19 ± 1.24	7.9 ± 0.51	6.13 ± 0.75	0.000 ^c^
HG strength/kg		23.32 ± 17.7	33.9 ± 17.27	14.8 ± 13.10	0.000 ^c^	29.2 ± 9.34	31.86 ± 8.9	28.88 ± 0.02	0.129 ^c^
Calf affected		36.02 ± 4.7	37.96 ± 2.90	31.23 ± 2.17	0.000 ^c^	35.7 ± 3.40	37.84 ± 2.2	32.68 ± 4.30	0.000 ^c^
Calf unaffected		36.3 ± 3.9	38.21 ± 2.80	31.53 ± 2.10	0.000 ^c^	35.56 ± 3.40	37.65 ± 1.9	31.18 ± 2.98	0.000 ^c^
CCI, median IQR		4(3–5)	4(3–4)	5(4–6)	0.000 ^b^	4(3–5)	3(3–5)	6(3–5)	0.177 ^b^
SARC-F, median, IQR		4(0–8)	2(0–4)	8(5–10)	0.000 ^b^	4(2–8)	4(2–8)	8(2–8)	0.343 ^b^
NIHSS, median, IQR		5(3–9)	4(3–6)	8(6–14)	0.000 ^b^	6(4–8)	6(4–8)	10(4–9)	0.961 ^b^

DM: diabetes mellitus, HTN: hypertension, IHD: ischemic heart disease, BMI: body mass index, SMI: skeletal muscle index, FIM: functional independence measure, NIHSS: National Institute of Health Stroke Scale, CCI: Carlson comorbidity index, ^a^—Chi-square test, ^b^—Manne–Whitney U test,^c^—*t*-test.

**Table 2 healthcare-10-02336-t002:** The basic characteristic of stroke survivors between the Egyptian versus Chinese female patients with or without sarcopenia.

Variables		Stroke Survivors in Egyptian Patients				Stroke Survivors in Chinese Patients			
		Total, n (%)	Without Sarcopenic	With Sarcopenic	*p*-Value	Total	Without Sarcopenic	With Sarcopenic	*p*-Value
Gender		200				195			
	Males	125(62.50)				125(64.10)			
	Females	75(37.50)	49	26		70(35.90)	60	10	
Age mean ± SD		55.54 ± 13.1	56.5 ± 14.1	57.1 ± 15.6	0.857 ^c^	52.8 ± 15.15	53.4 ± 15.15	45 ± 13.7	0.143 ^c^
Modified Rankin median, IQR			3(1–4)	4(1–4)	0.763 ^b^	4(3–4)	4(3–4)	4(3–4)	0.467 ^b^
Stroke type, n (%)									
	Ischemic	121(60.50)	26(53.10)	21(80.8%)	0.018 ^a^	84(43.10)	25(41.7%)	8(80%)	0.193 ^a^
	Hemorrhagic	79(39.50)	23(46.9)	5(19.2%)		111(56.9)	35(58.3%)	2(20%)	
Affected side, n (%)									
	Right side Left side	102(51) 98(49)	20(40.8%) 29(59.2%)	13(50%) 13(50%)	0.446 ^a^	66(32.80) 129(66.20)	16(26.6%) 44(73.3%)	2(20%) 8(80%)	0.438 ^a^
Smoking n (%)									
	No	110(55)	49(100%)	25(96.2%)	0.167 ^a^	120(61.50)	45(75%)	7 (70%)	0.738 ^a^
	Yes	90(45)	0(0%)	1(3.8%)		75(38.50)	15(25%)	3(30%)	
Drinking history, n (%)									
	No	200(100%)	46(100%)	29(100%)	------	99(50.70)	45(75%)	4(60%)	0.323 ^a^
	Yes	0(0.0)	0(0.0)	0(0%)		96(49.20)	15(25%)	6(40%)	
HTN, n (%)									
	No	29(14.50)	9(18.4%)	3(11.5%)	0.443 ^a^	49(24.70)	7(11.7%)	1(10%)	0.785 ^a^
	Yes	171(85.5%)	40(81.6%)	23(88.5%)		146(73.7%)	53(88.7%)	9(90%)	
DM, n (%)									
	No	118(59)	25(51%)	13(50%)	0.933 ^a^	24(12.30)	21(35%)	3(30%)	0.878 ^a^
	Yes	82(41)	24(49%)	13(50%)		171(87.70)	39(65%)	7(70%)	
IHD, n (%)									
	No	142(71)	33(67.3%)	16(61.5%)	0.615 ^a^	165(84.60)	55(91.7%)	90(90%)	0.862 ^a^
	Yes	58(29)	16(32.7%)	10(38.5%)		30(15.40)	5(8.6%)	1(10%)	
Performed conventional rehabilitation in the early phase, n (%)									
	No	116(58)	30(61.2%)	18(69.2%)	0.492 ^a^	65(33.30)	15(25%)	8(80%)	0.001 ^a^
	Yes	84(42)	19(38.8%)	8(30.8%)		130(66.70)	45(75%)	2(20%)	
Dyslipidemia									
	NO	140(70)	35(71.4%)	17(65.4%)	0.589 ^a^	72(36.90)	37(63.8%)	9(75%)	0.064 ^a^
	Yes	60(30)	14(28.6%)	10(34.6%)		123(63.10)	21(36.2%)	3(25%)	
Mini Nutritional Assessment-Short Form(MNA-SF)									
	Malnutrition	59(29.50)	18(36.7%)	13(46.2%)	0.727 ^a^	65(33.30)	23(38.3%)	8(80%)	0.027 ^a^
	Risky	59(29.50)	15(30.6%)	7(26.9%)		45(23.07)	14(23.3%)	2(20%)	
	Normal	82(41.0)	15(32.7%)	7(26.9%)		85(43.50)	23(38.3%)	0	
FIM, median, IQR									
	FIM motor	48(45–72)	60(41–71)	42(42–71)	0.683 ^b^	55(40–66)	55(39–70)	40(40–60)	0.620 ^b^
	FIM cognitive,	31(28–33)	31(29–32)	30(27–34)	0.884 ^b^	31(29–33)	31(28–34)	28(29–32)	0.819 ^b^
	FIM total	81(72–105)	60(71–103)	78(69–104)	0.781 ^b^	62(69–100)	65(65–105)	47(70–92)	0.559 ^b^
Time from stroke onset per month, median, IQR		57(34–43)	38(33–42)	35(34–44)	0.492 ^b^	49(35–44)	43(33–43)	34(33–37)	0.485 ^b^
Weight /KG		71.62 ± 11.0	66.5 ± 10.5	68.12 ± 12.38	0.493 ^c^	69.9 ± 12.12	60.3 ± 9.1	59. 8 ± 4.8	0.886 ^c^
Height/M		167.60 ± 8.1	159.1 ± 5.9	162.5 ± 7.2	0.036 ^c^	170.0 ± 7.26	161.7 ± 5.5	166.62 ± 3.7	0.010 ^c^
BMI (kg/m2)		25.3 ± 4.20	25.4 ± 4.2	25.5 ± 5.7	0.883 ^c^	24.0 ± 3.04	22.9 ± 2.7	21.5 ± 1.9	0.123 ^c^
SMI (kg/m2)		7.1 ± 1.50	6.5 ± 0.8	4.4 ± 0.56	0.000 ^c^	7.19 ± 1.24	6.8 ± 0.78	4.6 ± 0.57	0.000 ^c^
HG strength /kg		23.32 ± 17.7	12.2 ± 9.1	12.2 ± 10.5	0.995 ^c^	29.2 ± 9.34	27.3 ± 8.3	14.8 ± 0.421	0.000 ^c^
Calf affected		36.02 ± 4.7	37.6 ± 3.1	30.2 ± 6.5	0.000 ^c^	35.7 ± 3.40	34.6 ± 1.5	29.7 ± 2.34	0.000 ^c^
Calf unaffected		36.3 ± 3.9	37.8 ± 3.1	31.4 ± 2.1	0.000 ^c^	35.56 ± 3.40	35.2 ± 2.01	29.1 ± 2.22	0.000 ^c^
CCI, median IQR		4(3–5)	4(2–5)	6(3–6)	0.069 ^b^	4(3–5)	4(3–5)	5(4–8)	0.000 ^b^
SARC-F, median, IQR		4(0–8)	4(2–8)	10(3–8)	0.567 ^b^	4(2–8)	4(3–8)	6(7–8)	0.000 ^b^
NIHSS, median, IQR		5(3–9)	5(4–10)	10(4–10)	0.567 ^b^	6(4–8)	5(4–7)	9(8–15)	0.000 ^b^

DM: diabetes mellitus, HTN: hypertension, IHD: ischemic heart disease, BMI: body mass index, SMI: skeletal muscle index, FIM: functional independence measure, NIHSS: National Institute of Health Stroke Scale, CCI: Carlson comorbidity index, ^a^—Chi-square test, ^b^—Manne–Whitney U test, ^c^—*t*-test.

**Table 3 healthcare-10-02336-t003:** The Spearman correlation between pre- and post-stroke contributing variables that increase or decrease HGS and SMI among Egyptian and Chinese populations.

Pre-Stroke Variables								Post-Stroke Variables				
Variables	Age	Smoking	Drinking	HTN	DM	IHD	Dyslipidemia	Rehabilitation	FIM Motor	FIM Cognitive	MNA-SF	NIHSS
HG in Egyptian	−0.405 **	0.294 **	-----	−0.058	−0.450 **	−0.375 **	−0.363 **	0.037	0.568 **	0.455 **	0.624 **	−0.664 **
HG in Chinese	−0.222 **	0.280 **	0.229 **	0.003	−0.094	−0.003	−0.254 **	0.153*	0.233 **	0.173 *	0.583 **	−0.173 *
SMI in Egyptian	−0.303 **	0.458 **	----	−0.110	−0.173 **	−0.184 **	−0.063	0.110	0.171 *	0.198 **	0.298 **	−0.268 **
SMI in Chinese	0.030	0.161 *	0.169 *	0.053	−0.015	0.087	−0.148*	0.022	0.111	0.046	0.215 **	−0.048

** Correlation is significant at the 0.01 level (2-tailed). * Correlation is significant at the 0.05 level (2-tailed). HTN (hypertension), DM (diabetes mellitus), IHD (ischemic heart disease), SMI (skeletal muscle index), HG (hand grip), FIM (functional independence measure), MNA-SF (Mini Nutritional Assessment-Short Form), NIHSS (National Institute of Health Stroke Scale).

**Table 4 healthcare-10-02336-t004:** The multivariate regression stepwise analysis results to identify pre- and post-stroke variables of possible sarcopenia and sarcopenia in both populations.

Variables	HGS in Egyptian	HGS In Chineses	SMI in Egyptian	SMI in Chinese
	ꞵ	ꞵ	ꞵ	ꞵ
-Age Sig CI	−0.250 0.000 (−0.499)–(−0.175)	−0.290 0.000 (−0.258)–(−0.100)	−0.271 0.000 (−0.047)–(−0.018)	
*-Smoking history* Sig CI	0.303 0.0004 (6.906)–(14.634)	0.293 0.000 (3.133)–(8.085)	0.462 0.000 (1.080)–(1.831)	
-Drinking History Sig CI				
-HTN Sig CI				
-DM Sig CI	−0.198 0.002 (−11.579)–(−2.624)			
-IHD Sig CI	−0.192 0.001 (−12.003)-(−2.940)			
-Dyslipidemia Sig CI	−0.213 0.000 (−12.736)–(−3.698)	−0.225 0.001 (−6.832)–(−1.884)		
-Rehabilitation Sig CI			0.147 0.032 (0.042)–(0.892)	
-MNA-SF Sig CI	0.319 0.000 (3.659)–(9.884)	0.586 0.000 (5.048)–(7.511)	0.303 0.000 (0.319)–(0.823)	0.244 0.000 (0.137)–(0.491)
-FIM Cognitive				
FIM Motor				
***-NIHSS*** Sig CI	−0.408 0.000 (−2.685)–(−1.266)			

HTN (hypertension), DM (diabetes mellitus), IHD (ischemic heart disease), SMI (skeletal muscle index), MNA-SF (Mini Nutritional Assessment-Short Form), HGS (hand grip strength), FIM (functional independence measure), NIHSS (National Institute of Health Stroke Scale). Confidence interval (CI at 95%), Sig (significance).

## Data Availability

Data of the present study can be made applicable upon reasonable request to the corresponding author.

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
