# Peer review of "Stroke-Related Sarcopenia among Two Different Developing Countries with Diverse Ethnic Backgrounds (Cross-National Study in Egypt and China)"

_healthcare, 2022, doi:10.3390/healthcare10112336_

Round 1

Reviewer 1 Report

1. Please write with no italics, lines 12-14.

2. Please insert the references before the dot. As you did in the previous sentence.

3. Please capitalize the (These) line 48.

4. Please provide a more clear flowchart. I can not see the patients were excluded (Figure 1).

5. Please provide your results with bars or pies. 

Reviewer 2 Report

The authors studied the prevalence of sarcopenia and their risk factors in 195 Chinese patients and in 200 Egyptian stroke patients. They found a prevalence from 20% to 34.4%; in addition, they identified several risk factors, including: age, history of dyslipidemia, diabetes mellitus, ischemic heart disease and stroke severity

The article is interesting and well written. Below my comments.

First of all it should be considered that the authors performed a cross-sectional study with only one assessment. Therefore, the authors studied the prevalence and not the incidence. Sometimes, the authors use the term incidence in the text instead of prevalence. I recommend always using the term prevalence.

The authors state to have studied the risk factors for sarcopenia. To study the risk factors, it is necessary to perform a longitudinal study with two assessments and make sure that the exposure (risk factor) really precedes the outcome. For example, how are the authors sure that ante dyslipidemia actually preceded the sarcopenia? Sarcopenia may have appeared first and dyslipidemia later. In this case, it cannot be stated with absolute certainty that dyslipidemia is a risk factor for the onset of sarcopenia. I would recommend talking about variables (and not risk factors) that influence the onset of sarcopenia.

How were the variables used in the study collected? During a follow-up visit? By whom were they collected? Physiotherapist? Physicians? Nurse?

Reliability values of the measuring instruments should be provided

Reviewer 3 Report

The present study aimed to evaluate the prevalence of possible sarcopenia and sarcopenia among Egyptian and Chinese stroke survivors using the Asian working group of sarcopenia (AWGS-2019) and assess the factors associated with the incidence rate.

This is of interest in order to understand the needs of this population and to be able to implement preventive and therapeutic measures.

However, some questions or suggestions are raised below.

1.       It is necessary to include the validation information of the AWGS-2019 instrument (line 166).

2.       Some information should go in other sections, such as the following that corresponds to results section rather than to statistical methods section: “A total of 395 participants, […] 64.14% Chinese.”

3.       In the Statistical methods section, there seems to be an error when it is said that chi-square test is used for non-parametric variables instead of categorical variables.

4.       It would be advisable to add a calculation of the sample size of the study.

5.       It seems that the sample in the participants with sarcopenia is small, especially the Chinese participants with sarcopenia (n = 10), which could condition the results of the multivariate regression stepwise analysis.

Other Suggestions:

1.       Attention should be paid to the formatting of the article, as there are some errors. For example, below table 3 is the title of table 4 and the notes below the table are omitted. In the use of references, sometimes they are placed before the full stop, sometimes after, even in line 291 the name of the authors is omitted, among others. Some capital letters are also missing after the full stop. Finally, Figure 1 has been moved and the boxes cannot be read.

Author Response

Please see the the attachment

Round 2

Reviewer 2 Report

The authors satisfied all my requests

According to me, this version of tha manuscript can be accepted

Reviewer 3 Report

Thank you for addressing the suggestions from the previous review.